# Canine Oral Melanoma Genomic and Transcriptomic Study Defines Two Molecular Subgroups with Different Therapeutical Targets

**DOI:** 10.3390/cancers14020276

**Published:** 2022-01-06

**Authors:** Anais Prouteau, Stephanie Mottier, Aline Primot, Edouard Cadieu, Laura Bachelot, Nadine Botherel, Florian Cabillic, Armel Houel, Laurence Cornevin, Camille Kergal, Sébastien Corre, Jérôme Abadie, Christophe Hitte, David Gilot, Kerstin Lindblad-Toh, Catherine André, Thomas Derrien, Benoit Hedan

**Affiliations:** 1IGDR—UMR 6290, CNRS, University of Rennes 1, 35000 Rennes, France; prouteauanais@gmail.com (A.P.); stephanie.mottier@univ-rennes1.fr (S.M.); aline.primot@free.fr (A.P.); edouard.cadieu@univ-rennes1.fr (E.C.); laura.bachelot@univ-rennes1.fr (L.B.); nadine.botherel@univ-rennes1.fr (N.B.); armel.houel@univ-rennes1.fr (A.H.); camille.kergal@univ-rennes1.fr (C.K.); sebastien.corre@univ-rennes1.fr (S.C.); christophe.hitte@univ-rennes1.fr (C.H.); david.gilot@univ-rennes1.fr (D.G.); catherine.andre@univ-rennes1.fr (C.A.); 2Laboratoire de Cytogénétique et Biologie Cellulaire, CHU de Rennes, INSERM, INRA, University of Rennes 1, Nutrition Metabolisms and Cancer, 35000 Rennes, France; florian.cabillic@chu-rennes.fr (F.C.); laurence.cornevin@univ-rennes1.fr (L.C.); 3Laboniris, Department of Biology, Pathology and Food Sciences, Oniris, 44300 Nantes, France; jerome.abadie@oniris-nantes.fr; 4Broad Institute of MIT and Harvard, Cambridge, MA 02142, USA; kersli@broadinstitute.org; 5Science for Life Laboratory, Department of Medical Biochemistry and Microbiology, Uppsala University, SE-751 24 Uppsala, Sweden

**Keywords:** mucosal melanoma, dog model, oncogenes, immune checkpoint inhibitors, chromosomal rearrangements, *CDK4*, *MDM2*

## Abstract

**Simple Summary:**

In humans, mucosal melanoma (MM) is a rare and aggressive cancer. The canine model is frequently and spontaneously affected by MM, thus facilitating the collection of samples and the study of its genetic bases. Thanks to an integrative genomic and transcriptomic analysis of 32 canine MM samples, we identified two molecular subgroups of MM with a different microenvironment and structural variant (SV) content. We demonstrated that SVs are associated with recurrently amplified regions, and identified new candidate oncogenes (*TRPM7*, *GABPB1*, and *SPPL2A*) for MM. Our findings suggest the existence of two MM molecular subgroups that could benefit from dedicated therapies, such as immune checkpoint inhibitors or targeted therapies, for both human and veterinary medicine.

**Abstract:**

Mucosal melanoma (MM) is a rare, aggressive clinical cancer. Despite recent advances in genetics and treatment, the prognosis of MM remains poor. Canine MM offers a relevant spontaneous and immunocompetent model to decipher the genetic bases and explore treatments for MM. We performed an integrative genomic and transcriptomic analysis of 32 canine MM samples, which identified two molecular subgroups with a different microenvironment and structural variant (SV) content. The overexpression of genes related to the microenvironment and T-cell response was associated with tumors harboring a lower content of SVs, whereas the overexpression of pigmentation-related pathways and oncogenes, such as *TERT*, was associated with a high SV burden. Using whole-genome sequencing, we showed that focal amplifications characterized complex chromosomal rearrangements targeting oncogenes, such as *MDM2* or *CDK4*, and a recurrently amplified region on canine chromosome 30. We also demonstrated that the genes *TRPM7*, *GABPB1*, and *SPPL2A*, located in this CFA30 region, play a role in cell proliferation, and thus, may be considered as new candidate oncogenes for human MM. Our findings suggest the existence of two MM molecular subgroups that may benefit from dedicated therapies, such as immune checkpoint inhibitors or targeted therapies, for both human and veterinary medicine.

## 1. Introduction

Melanoma is the deadliest skin cancer in humans, with an incidence of 80,000 cases per year in the USA [1]. Mucosal melanoma (MM), a rare clinical entity, accounts for 1–2% [2,3] of all melanomas; however, this rate is approximately 22% in certain Asian countries where cutaneous melanoma (CM) occurs less frequently [4]. MM is caused by melanocytes that reside in the mucous membranes of the respiratory, gastrointestinal, and urogenital tracts, and occurs mainly in the head and neck (31–55%), anorectal (17–24%), and vulvovaginal (18–40%) regions [3]. Compared to CM, MM is highly aggressive and has a less favorable prognosis [2,5,6], with a 5-year survival rate of 20–35% based on the disease location and stage [2,3,6,7]. Treatment options include surgical resection and/or radiation therapy to achieve locoregional control; however, the prognosis remains poor. Some advanced cases of MM may benefit from targeted therapy, such as the usage of KIT inhibitors or immunotherapy using checkpoint inhibitors [2,3,8,9,10]. However, the response to immunotherapy is highly variable, and patients with MM have a lower response rate than those with CM [3,8].

In recent years, large genomic studies involving whole exome sequencing (WES) and, more recently, whole genome sequencing (WGS) have been conducted to examine the mechanisms of MM tumor initiation, progression, and metastasis, and to find more suitable therapies [11,12,13,14,15,16,17,18]. These studies have shown that MM presents a lower mutation burden and has a greater load of structural variants (SVs) and copy number alterations (CNAs) than CM [11,12,14,15]. In MM, SV and CNA frequently involve known oncogenes, such as *CDK4*, *MDM2*, and *TERT* [14,15,16,18]. Two recent studies have identified two MM subgroups based on the pattern of complex clustered SVs [14,15]. In oral melanomas comprising one such pattern, some of the SVs were linked to poor outcomes [15]. Since MM is a rare entity in humans, popular breeds of dog patients with MM can be evaluated to better understand the molecular characteristics of these MM subgroups and to reveal relevant therapeutic targets.

In the last decade, dog models have emerged as being unique, spontaneous, and immunocompetent for human cancers [19,20,21,22,23,24], particularly for MM [13,22,23]. Canine MM is the most frequently occurring oral malignancy in dogs and shares several clinical, biological, and genetic features with its human counterpart. As in humans, canine MM tends to rapidly metastasize to distant organs and responds poorly to chemotherapy. Immunotherapy in dogs has been developed to create a melanoma vaccine; however, the results remain controversial, with highly variable responses [23,25,26]. Recently, the development of anti PD-L1 [27,28] and anti-PD1 antibodies [26] for canine patients with MM has proven to be effective in pilot studies. Genetically, canine MM mimics human MM with a predominance of somatic CNAs and complex SVs, and a relatively low tumor mutation burden [13,24,29]. At the transcriptomic level, only a few studies in canine MM tumors involving a limited number of samples have identified deregulation in multiple pathways, such as MAPK/ERK, PI3K/AKT, “cytokine receptor interaction,” “ECM receptor interaction”, and “focal adhesion” [30,31,32].

In this study, we examined the genetic basis of MM in a spontaneous canine model in specific breeds that are frequently affected by MM by combining genomic and transcriptomic sequencing experiments together with functional validations. Unsupervised clustering analysis of RNAseq data from 32 canine MM samples highlighted the existence of two molecular subgroups of MM with differential expressions: one characterized by immune and microenvironment signatures, and the second by the overexpression of oncogenes (such as *MDM2*, *CDK4*, and *TERT*, which were characterized by the enrichment of complex chromosomal rearrangements). The effect on cell proliferation of genes, which is frequently altered through chromosomal rearrangements, was explored in both canine and human MM cell lines. The results of the study suggest the existence of two MM molecular subgroups that could benefit from distinct therapies and lead to further translational studies in human and veterinary oncology.

## 2. Materials and Methods

### 2.1. Sample Collection

Blood and tissue biopsy samples from dogs with oral melanoma were collected for this study through Cani-DNA (http://dog-genetics.genouest.org, accessed on 18 November 2021) and the Canine Comparative Oncology Genomics Consortium Biological Resource Centers BRC (www.ccogc.org, accessed on 1 July 2021) (61 and 18 cases, respectively) (Appendix A). Oral melanoma diagnoses were confirmed based on the histopathological analysis (JA) with adequate immunostaining (expression of the S100, MelanA, PNL2 markers). DNA and RNA were extracted, as previously described [21,33]. Blood and tissue sampling was performed by veterinarians on privately owned dogs during the course of the health follow-up, with the owner’s consent.

### 2.2. Canine Cell Lines

The Bear (accession numbers CVCL_OD14) and CML10 (accession number CVCL_IZ11) cell lines were obtained from Dr. J. Modiano (Colorado State University, Fort collins, CO, USA). The cell lines were cultured at 37 °C in RPMI 1640 medium (Gibco, Amarillo, TX, USA), supplemented with 10% fetal bovine serum and 0.2% primocin (Invivogen, San Diego, CA, USA). Five other cell lines, obtained from fresh canine oral melanoma samples (Dog-OralMel-18249, Dog-OralMel-18395, Dog-OralMel-18333, Dog-OralMel-18848, and Dog-OralMel-18657), were developed in the laboratory (Appendix A). These cell lines were cultured in DMEM/F-12 medium (Gibco), supplemented with 10% fetal bovine serum and 0.2% primocin. After ten passages, RPMI 1640 was used. All cell lines were tested for mycoplasma using the MycoAlertTM Plus kit (Lonza, Rockland, ME, USA), and were found to be mycoplasma-free.

### 2.3. Fluorescence In Situ Hybridization

Fluorescence in situ hybridization analyses were performed on chromosome preparations generated from the cell lines using the following conventional techniques: colcemid arrest, hypotonic treatment, and methanol/glacial acetic acid fixation, as described previously [34]. The following bacterial artificial chromosome (BAC) clones were used: CH82-199H02 and CH82-179B09 for the *MDM2* region, CH82-213B06 and CH82-204K11 for the *CDK4* region, CH82-99P23 and CH82-60O16 for the CFA 7 region (58.4 Mb to 58.9 Mb), CH82-1E17 and CH82-40I15 for the region containing *GABPBP1*, and *USP8* and *TRPM7* amplifications (https://bacpacresources.org/, accessed on 1 July 2021). These BAC clones were labeled using green-dUTP (Abbott Molecular, Des Plaines, IL, USA) and Cy3-dCTP (Amersham Biosciences, Chalfont, UK). The slides were analyzed by an experienced cytogeneticist (FC) using a fluorescence microscope (Axioskop2, Axio Imager Z2, Zeiss, Göttingen, Germany) and Isis imaging software (Metasystems, Altlussheim, Germany). At least 100 non-overlapping tumor nuclei were examined in this study.

### 2.4. RNAseq Clustering and Signature Analysis

For the 32 canine MM samples, polyadenylated RNAs were extracted, sequenced, and analyzed, as previously described [33]. All RNAseq fastq files are available in the SRA under the BioProject PRJNA749900. Briefly, the pipeline used the “canFam3.1-plus” annotation as the reference annotation [35], and the canFam3.1 assembly version as the reference genome [36]. Based on the protocol described by Djebali et al. [37], FASTQ reads were aligned to the transcriptome and genome using the STAR program (v2.5.0a) [38]. Finally, gene expression levels were estimated as raw counts (unnormalized) using the RSEM program (v1.2.25) [39] for each sample individually, and subsequently merged to obtain a matrix expression file (with genes in rows and samples in columns). This matrix of expression was then normalized and transformed across all samples in order to stabilize the variance using the DESEq2 (v1.22.2) [40] function *vstcounts* with the option “blind = TRUE”.

To select the genes that would provide the most information for clustering analysis, we used 6000 of the most variable genes, that is, those with the highest median absolute deviation. Then, the nonnegative matrix factorization (NMF) algorithm from the NMF R package (v0.22.0) [41] was applied for different values of *k* clusters/ranks (*k* = 2–6). To determine the best *k* cluster, the *nmfEstimateRank* function was used (Appendix B Figure A1), and the consensus matrix method (Silhouette) identified *k* = 2 as the optimal number of clusters. Finally, we employed the NMF function *extractFeatures* to obtain cluster-specific gene signatures for the two clusters (Appendix A). These gene lists were then used as the input for the gprofiler2 (v0.1.9) R program [42] with the “*gost*” function, with the “organism” set to “cfamiliaris” to perform a gene set enrichment analysis over multiple databases (Gene Ontology, KEGG, CORUM). Finally, heatmaps were plotted using the complexHeatmap software (v2.3.1) [43] to integrate and visualize multiple sources of information (gene signatures, sample clustering, SV content, and oncogene mutational status) (https://github.com/tderrien/Prouteau_et_al, accessed on 18 November 2021).

### 2.5. Whole Genome Sequencing

DNA from four canine oral melanoma cell lines and the blood samples of two affected dogs were extracted for whole genome sequencing (WGS), as previously described [21]. WGS was performed with the BGI sequencing platform (BGI, Shenzhen, China), using BGISEQ-500 short-read sequencing as previously described [44]. Briefly, 1000 ng of genomic DNA was quantified using a Qubit 3.0 fluorometer (Life Technologies, Paisley, UK) and sheared using an E220 Covaris instrument (Covaris Inc., Woburn, MA, USA). Sizes were selected using a Vahstm DNA Clean beads kit (Vazyme, Nanjing, China) to an average size of 200–400 bp. The selected fragments were end repaired, and 3′ adenylated and BGISEQ-500 platform-specific adaptors were ligated to the A-tailed fragments. The ligated fragments were purified and amplified by PCR. Finally, circularization was performed to generate single-stranded DNA circles. After quantification, the libraries were loaded onto a sequencing flow cell and processed for 100 bp paired-end sequencing on the BGISEQ-500 platform.

### 2.6. Low Pass Sequencing

Low-pass sequencing of 42 formalin-fixed paraffin-embedded (FFPE) samples of melanoma was performed by Psomagen (Rockville, MD, USA). Briefly, the input DNA quality was verified using gel electrophoresis, and the quantity was measured using the Picogreen assay (Thermo Scientific, Waltham, MA, USA); 2 ng of DNA was prepared in 30 ul of buffer and used for library construction using the Nextera DNA Flex Library Kit (Illumina, San Diego, CA, USA), according to the manufacturer’s guidelines. The size of the final DNA libraries was then validated using the TapeStation D1000 ScreenTape (Agilent, Santa Clara, CA, USA) and D1000 reagents (Agilent, Santa Clara, CA, USA). The quantity was measured using the Picogreen assay (Thermo Scientific, Waltham, MA, USA), and the molar concentration was calculated using both sources. The libraries were normalized to 2 nM, pooled in equimolar volume, and then loaded on the flow cell from the Novaseq S4 300 cycle kit (Illumina, San Diego, CA, USA) and the XP-4lane kit (Illumina, San Diego, CA, USA). The prepared flow cell and SBS cartridge from the Novaseq S4 300 cycle kit (Illumina, San Diego, CA, USA) were inserted into the Novaseq 6000 system and sequenced using 151-10-10-151 running parameters, reaching a mean depth of 0.48 X.

### 2.7. Mapping

After sequencing, the raw reads were filtered (adapter sequences, contamination, and low-quality reads were removed) according to the manufacturer’s guidelines. Sequence data were then aligned to the dog reference genome (assembly version canFam3.1) using BWA-MEM (version 0.7.17) [45], and PCR duplicate reads were removed using Picard tools (version 2.18.23) (http://broadinstitute.github.io/picard/, accessed on 1 July 2021). The read data were processed according to the GATK best practices; specifically, the base quality score recalibration was assessed using GATK4 (version 4.0.12).

### 2.8. Somatic Variant Calling

The Mutect2 tool from the GATK4 software was then used to call somatic SNVs and short INDELS against a panel of normal (PON), comprising matched normal samples, and against germline variants from 722 dog genomes (https://data.broadinstitute.org/vgb/Ostrander_VCFs/722g.990.SNP.INDEL.chrAll.vcf.gz, accessed on 1 July 2021). Variant annotation was performed using the VEP program [46] with the EnsEMBL “Canis familiaris” annotation (v. 95).

### 2.9. Somatic Copy Number Variant Calling

The somatic copy number variants (CNVs) were determined using the “R” Package DNAcopy [47]. The canine genome was split into bins with a window size of 10 kb, or based on the exome target regions for WGS and WES. The number of reads was normalized to the total number of reads per sample. The log2 ratio with the corresponding germinal DNA, when available (or with a pool of three normal DNA for BEAR or CML10 cell lines), was estimated for each window. Several windows were merged (segmentation) into a larger segment with a significance threshold of 1 × 10^4^, and the copy number of the segments was estimated.

### 2.10. SV Calling

Based on the aligned bam files from the BWA-MEM software, four structural variant types were identified: breakends (BND), deletions (DEL), duplications (DUP), and inversions (INV) using Delly software [48] (version v0.8.3). Only somatic SVs with the flags “PASS” and “PRECISE” were retained. Finally, circular representations of the SVs and the gain/loss from the DNA copies along the canine chromosomes were measured using the circlize R package [49] (version 0.4.10) (https://github.com/tderrien/Prouteau_et_al, accessed on 18 November 2021).

### 2.11. Modeling Chromothripsis Events

To assess the patterns of chromothripsis, we tested for the enrichment of chromosome-specific SVs by modeling the association between SV breakpoints and chromosome length using hypergeometric distributions (or a hypermetric distribution). More precisely, we defined “*x*” as the number of SV breakpoints on a specific chromosome out of a total of “*k*”-detected breakpoints, where “*m*” was the chromosome size and “*N*” was the size of the canine genome. The computed *p*-value was the probability of “*x*” SVs in the total number of SVs, calculated using dhyper (*x = x, m = m, n = N − m, k = k*) in R. Statistical significance was set at *p* < 10^−3^. The clustering of the SV breakpoints was tested as previously described [50]. Chromothripsis regions were detected with Shatterproof software [51], using the copy number and BND information along with the standard parameters.

### 2.12. Classification of Dogs with “Low” versus “High” Content of Structural Variants (SV)

High/low SV classification was defined using the copy number status of segments resulting from the WES data. SV samples were defined as “high” if they had an abundance of copy number states oscillating on at least one chromosome (copy number states ≥10 states) [14], with at least one major amplification (copy number ≥ 90 percentile of whole genome amplifications). This classification was verified using manual curation.

### 2.13. Targeted Copy Number and Expression Analysis

High-throughput quantitative PCR (qPCR) was performed using the SmartChip system (Takara Bio, Kusatsu, Japan) on 47 oral melanoma samples to detect focal amplifications on tumor DNA on CFA 10 and CFA 30. We used primer pairs targeting two genes on CFA 10 (*MDM2* and *CDK4*) and four genes on CFA 30 (*TRPM7*, *USP8*, *SPPL2A*, and *GABPB1*) (Appendix A). A primer pair targeting a region of CFA 9 was used as an internal control, and each experiment was performed using DNA from an unaffected dog as an external control. qPCR was performed on tumor DNA samples after pre-amplification using the SYBR green PCR master mix (Thermo Fisher Scientific, Waltham, MA, USA) on the 7900HT Fast Real-Time PCR System (Applied Biosystems, Waltham, MA, USA), using standard procedures. Each sample was measured thrice, and the relative amounts of the sequences were determined using the ΔΔCt method.

RNA was extracted from the 47 tumor samples and reverse transcription was performed on 1 μg of RNA from tumorous or healthy tissue using the High-Capacity cDNA Reverse Transcription Kit (Applied Biosystems, Waltham, MA, USA), according to the manufacturer’s instructions. RT-qPCR was performed using the same target genes. *SAP130* was used as the control housekeeping gene, and a pool of data from non-affected oral tissues of four dogs was used as the external control. Each sample was measured thrice, and the relative amounts of the sequences were determined using the ΔΔCt method.

### 2.14. Western Blot

Cellular protein extracts from cell lines were prepared using a cell lysis buffer containing 18 mmol/L Tris-HCl, pH 7.5; 135 mmol/L NaCl, 0.9 mmol/L EDTA; 0.9 mmol/L EGTA and supplemented with 1 mmol/L PMSF; 1× EDTA-free cocktail protease inhibitor (Roche Diagnostics, Bale, SWISS); 30 mmol/L sodium fluoride, 40 mmol/L glycero phosphate; 1 mmol/L sodium orthovanadate; and 0.5% Triton X-100.

Protein concentrations were determined using the BCA protein assay (Sigma-Aldrich, St. Louis, MO, USA) with bovine serum albumin as a standard. Protein samples were denatured for 10 min at 95 °C, and equal amounts of cell proteins (50 μg) were subjected to 10% SDS-PAGE and transferred onto nitrocellulose membranes (Amersham-GEH Life, Buckinghamshire, UK). The membranes were probed with the appropriate antibodies. The primary antibodies used were: anti-CDK4 (559693, BD Pharmingen, San Diego, CA, USA), anti-MDM2 (clone 2A10, ref MABE281, Merckmillipore, Darmstadt, Germany), and anti–ERK (sc-94, Santa Cruz Biotechnology, Dallas, TX, USA). Horseradish peroxidase-conjugated secondary antibodies were purchased from Jackson ImmunoResearch (West Grove, PA, USA). Signals were detected using a LAS-4000 Imager (Fuji Photo Film, Tokyo, Japan).

### 2.15. Transfection with Specific and Control siRNAs and Cell Proliferation Assays

The siRNAs (80 nM) for CDK4, GABPB1, TRPM7, USP8, SPPL2A, and control siRNA (IDT) were transfected into the Bear cell line using Lipofectamine 2000 (Invitrogen, Waltham, MA, USA), according to the manufacturer’s recommendations (Appendix A). CDK4 siRNA was used as the positive control. The siRNAs (10 nM) for GABPB1, TRPM7, and SPPL2A, as well as the control siRNA (IDT), were transfected into two human cell lines (HMV-2/CVCL_1282 -Merck-, WM3211/CVCL_6797 -Rockland-) using Lipofectamine 2000 (Invitrogen, Waltham, MA, USA) following the manufacturer’s recommendations (Appendix A). Cells were seeded in 6-well plates in 1 mL of RPMI 1640 medium (Gibco, Amarillo, TX, USA), supplemented with 10% fetal bovine serum and 0.2% primocin (Invivogen, Waltham, MA, USA) at a density of 1 × 10^6^ cells, and incubated at 37 °C. The medium was changed 6 h post-transfection to avoid toxicity.

Cell proliferation was evaluated in 96-well plates, 72 h after transfection with an initial density of 30,000 cells per well, using a methylene blue colorimetric assay. Briefly, the cells were fixed for 30 min in 90% ethanol, removed, dried, and subsequently stained for 30 min using 1% methylene blue dye in borate buffer. The fixed cells were washed 4–5 times using tap water, and 100 µL of 0.1 N HCl was added to each well. Cell density was analyzed using a spectrophotometer at 620 nm.

## 3. Results

### 3.1. Transcriptome Landscape of Canine Oral Melanoma Highlights Two Distinct Subgroups

The ability of the expression profiles of the 32 canine MM tumors in predicting the prognostic subtypes of MM was first investigated (Appendix A). Using unsupervised clustering based on the nonnegative matrix factorization (NMF) algorithm (see Section 2), we observed that the canine tumor samples could be separated into two distinct clusters (group 1, 12 samples; group 2, 20 samples) based on the underlying structure of their expression data (Figure 1A). Since NMF organizes both the samples and genes, it also allows us to define a list of genes that represent the signatures of the two clusters (Appendix A). Using gene set enrichment analysis with the gprofiler tool [42], we showed that the 384 genes with the group 1 signature (Appendix A) were significantly associated with immune-related functional gene sets (Biological Process Term “Immune Response” *p* = 1.6 × 10^−24^, “cytokine-mediated signaling pathway” *p* = 2.05 × 10^−15^) (Appendix A). This suggests that the microenvironment differs between both groups, with the MM in group 1 presenting a higher degree of immune cell infiltration. This was confirmed by the overexpression of genes encoding components of the T-cell receptor (TCR) complex, such as *CD3E*, *CD3D*, *CD3G*, and *CD247* (Figure A2). Moreover, in this group, genes related to cytotoxic functions (such as granzyme B), the IFN-γ pathway (such as *IRF1*), or to the presence of an ongoing immune response and a cytokine-rich microenvironment (such as *IL18*, *CCL3*, *CCL4*, *CCL13*, and *CCL22*) [52] were significantly overexpressed (Figure A2). The ratio between immune cell types with an immunostimulatory or immunosuppressive function seemed to be more important than the absolute number of immune cell types in determining the antitumorigenic vs. protumorigenic role of the microenvironment [52]. Therefore, we determined the expression ratio of proinflammatory cytokines (IFN-γ, IL-1A, IL-1B, and IL-2), versus immunosuppressive molecules (IL-10, IL-11, IL-4, and TGFB1) in both groups. This ratio was significantly increased in group 1, suggesting a relative increase in pro-immunogenic responses in this group (*p* = 0.03, Student’s *t*-test; Figure 1B). This profile is concordant with that of “hot immune” tumors [53,54], and the overexpression of immune checkpoint genes such as *CTLA4* (log2fold change = 2.78; adjusted *p* = 1.22 × 10^−5^), *TIM3* (log2fold change = 1.6; adjusted *p*-value = 2.2 × 10^−9^), *LAG3* (log2fold change = 1.3; adjusted *p*-value = 5.8 × 10^−3^), or immunomodulating cytokines (including IL2RG). This suggests that this MM group may benefit from treatment with immune checkpoint inhibitors [53,54]. In addition, several studies have involved the phenotype switching of melanoma cells as an escape route to CM-targeted therapies using BRAF inhibitors [55,56]. Under the control of the microenvironment or intrinsic cell factors, melanoma cells can acquire a resistance to targeted therapies by switching from a proliferative to an invasive state [57,58]. These changes to an aggressive phenotype are associated with dedifferentiation (from a melanocytic/transitory to a neural crest-like/undifferentiated phenotype) and an epithelial-to-mesenchymal-like (EMT-like) transition that promotes metastasis [59]. Several genes associated with the acquisition of an invasive-dedifferentiated-EMT-like phenotype [57,60] were also overexpressed in group 1 (Appendix A).

In contrast, melanin metabolic processes and pigmentation pathways were identified in the second group (*n* = 20 samples), i.e., the biological process term “melanin biosynthetic process” *p* = 7.6 × 10^−4^, “melanocyte differentiation” *p* = 9.9 × 10^−3^, “pigmentation” *p* = 1.8 × 10^−4^) (Appendix A). Gene signatures from group 2 corresponded to a proliferative and differentiated phenotype of melanoma (*DCT*, *MLANA*, *TYR*) [57,61]. This was reflected by the overexpression of the melanocyte-specific transcription factor MITF (log2fold change = 2.2, adjusted *p*-value = 1.0 × 10^−5^) (Figure 1C), which is known to control the proliferation, migration, and invasion of melanoma cells in CMs. Interestingly, even if these CMs are highly proliferative, they are highly sensitive to targeted therapies, such as those using BRAF inhibitors [57]. This group presented with the overexpression of well-recognized cancer driver genes, such as *TERT* (log2fold change = 1.5, adjusted *p*-value = 0.028), *KIT* (log2fold change = 2.1, adjusted *p*-value = 5.0 × 10^−5^), or the oncogene *POM121* (log2fold change = 0.67, adjusted *p*-value = 0.011), which has recently been linked to poor prognosis in human oral MM [15] (Figure 1C).

SVs involving *MDM2* and *CDK4* genes are hallmarks of human and canine MM [14,15,16,18], and a recent study in humans pointed out two different MM subgroups based on SV profiling [14]. Using WES data of the corresponding canine MM, we analyzed their genomic SV profiles (Section 2) and classified canine tumors into “low SV” (*n* = 12) or “high SV” (*n* = 20) (Figure 1A) based on the distribution of SV features (number, intensity, and clustering). Interestingly, the two subgroups defined by the transcriptomic analysis significantly overlapped with the two groups defined according to SV status (exact Fisher test *p* = 9.5 × 10^−3^). The first transcriptome subgroup, that is, “group 1” in this study, which was characterized by an immune signature, comprised tumors with “low SV,” while the second transcriptome subgroup contained more “high SV” tumors. These results are concordant with previous studies in human cancers, showing that chromosomal instability and CNAs are associated with a decrease in cytotoxic immune cell infiltration [52] and resistance to checkpoint inhibitors [52,53,62]. Our findings suggest that oral melanomas behave similarly and may evolve through two different paths driven by SV somatic changes that would lead to marked differences in therapeutic options: the first group containing “hot immune” tumors with a lower numbers of SVs would likely benefit from immunotherapy, while the second group, which contains “cold immune” tumors with a higher number of SVs would respond better to therapies targeting amplified oncogenes, such as treatment using CDK4/6 inhibitors.

Given that SVs have been associated with resistance to immunotherapy in human cutaneous melanoma [53] and with a poor outcome in both human [15] and canine oral MM [24], we then focused on the characterization of these SVs in canine MM as models for therapies in human oral MM.

### 3.2. “High SV” MM Are Characterized by Focal Amplifications and Numerous Chromosomal Translocations

To refine the annotation of SVs, we performed WGS on four canine MM cell lines (Bear, CML10, Dog-OralMel-18333, and Dog-OralMel-18395 cell lines) at an average depth of 30X. Then, the Delly program [48] was used to catalog a total of 5906 SVs (DEL, DUP, INV, INS, and BND) on the four cell lines (Appendix A) (see Section 2). Three canine cell lines showed strong focal amplifications of CFA 30 and/or CFA 10, which are recurrently observed in canine MM [13,24,29,63,64] (Figure 2). The Bear cell line showed focal amplifications of the CFA 30, targeting the 16–17 Mb region (up to 67 copies) and of the CFA 10 targeting *MDM2* and *CDK4* region (up to 46 copies). The Dog-OralMel-18333 cell line was characterized by focal amplifications targeting the same region of CFA 30 (up to 27 copies) and CFA 7 (up to 13 copies), whereas the Dog-OralMel-18395 cell line presented strong focal amplifications on CFA 10 (up to 41 copies) targeting *MDM2* and *CDK4*. In contrast, the last cell line, CML10, did not show any of these alterations. These SV profiles suggest that the CML10 cell line belongs to MM group 1 while the three others to the group 2.

Interestingly, when analyzing the SV distribution in the tumor genome, we found that the regions harboring focal amplifications comprised significant amounts of several types of SVs, such as INVs, DELs, and insertions (Appendix A and Figure 2). In particular, these focal amplifications were found to display inter-or intra-chromosomal translocations with significant breakpoint clustering, similar to what is seen in human MM, particularly oral MM, which presents inter-and intra-chromosomal translocations between or within HSA 5 (*TERT*) and HSA 12 (*CDK4*) [14,16]. Thus, for the three canine MM cell lines with strong focal amplifications, the amplified regions presenting breakpoint clustering (see Methods) were detected as potential chromothripsis regions using the Shatterproof tool (Appendix A). For the three cell lines harboring these focal amplifications, inter-or intra-chromosomal translocations were explored using FISH.

It showed large genomic regions with repeated amplifications and fusions of CFA 30 and CFA 10 for the Bear, CFA 30 and CFA 7 for the Dog-OralMel-18333, and intra CFA10 for the Dog-OralMel-18395 cell lines (Figure 2C,F,I). Although we could not find the presence of double-minute chromosomes to explain the highly elevated copy numbers using FISH analysis, the three cell lines harboring strong focal amplifications presented features of chromothripsis, as defined by Korbel et al. [50], that is, breakpoint clustering and irregular oscillating copy number states.

Thus, the results of this study suggest that focal amplifications of the *CDK4* and *MDM2* genes, as well as those in the 16–17 Mb region of CFA 30 in canine MM, reflect major complex inter-and intra-chromosomal rearrangements. These massive DNA rearrangements, clustered with high amplifications and deletions, result in long derivative chromosomes arising from cataclysmic events compatible with chromothripsis [14,50]. These recurrent focal amplifications are expected to drive the oncogenic process of MM.

### 3.3. Exploring Candidate Oncogenes on CFA 30

#### 3.3.1. Correlation between Copy Numbers (CNA) and Expression of Candidate Oncogenes

Chromothripsis-like events often involve the amplification of oncogenes to promote tumor progression [65,66]. In canine MM, these recurrent amplifications target well-recognized oncogenes, such as *CDK4,* which are involved in the early phase of the cell cycle, and *MDM2,* whose protein inhibits p53, as well as several candidate oncogenes localized on CFA 30 amplifications, which remain unidentified in human MM. Based on the intersection of genomic intervals defining recurrent CFA30 amplifications in the WES data from 32 dogs, we highlighted the minimal recurrently amplified region, containing four genes that may be relevant candidate oncogenes: *TRPM7* is involved in the PI3K/AKT and MAPK pathways, as well as in EMT in ovarian and lung cancers [67,68,69,70]; *GABPB1* encodes the ß subunit of a transcription factor and is linked to cell proliferation and poor outcome in renal cell carcinoma [71]; *USP8* has been linked to tumorigenesis and poor prognosis in several cancers [72,73]; and *SPPL2A* plays a role in innate and adaptive immunity [74]. 

We hypothesized that the tumor advantages conferred by these focal amplifications were associated with the overexpression of the oncogenes targeted by these amplifications. Thus, we first checked whether the expression levels of the candidate oncogenes of CFA 30 (*TRPM7, USP8, GABPB1*, and *SPPL2A*) were correlated with the copy number of these genes. With this aim, we analyzed the copy numbers and gene expression of the candidate oncogenes in 47 novel canine MM samples by performing qPCR and RT-qPCR. A significant positive correlation between gene expression and copy number was found for all four CFA 30 candidate genes, with *SPPL2A* having the strongest correlation (Spearman coefficient: 0.76, *p* = 6.9 × 10^−9^). Concerning oncogenes located on CFA 10, the expression levels of *MDM2* and *CDK4* were positively correlated with DNA copy numbers (Spearman coefficient: 0.68, *p* = 8.9 × 10^−7^ and Spearman coefficient: 0.65, *p* = 1 × 10^−6^, respectively) (Figure 3A). Finally, Western blot analyses using cross-reactive available antibodies showed that the *MDM2* and *CDK4* focal amplifications were linked to a higher amount of protein (Figure 3B).

Thus, amplifications and chromosomal rearrangements of CFA 10 and CFA 30 led to a clear overexpression of the targeted genes and protein levels, suggesting the acquisition of new cell capacities in connection with the proliferative and aggressive features of MM cells.

#### 3.3.2. Effect of Candidate Oncogenes on MM Cell Proliferation

To evaluate the effect of the altered expression of the CFA30 genes on cancer cell proliferation, we assessed the proliferation capability of canine MM cancer cells after silencing *TRPM7, SPPL2A, GABPB1*, and *USP8* genes using colorimetric cell proliferation assays. Expression analysis showed that the siRNA experiments reduced the mRNA expression levels of the four oncogenes by 50–72% (Appendix A), and the silencing of the *TRPM7*, *SPPL2A*, and *GABPB1* genes significantly decreased cell proliferation from 18–28% (Figure 3C). These results suggest that the combined amplification of the CFA 30 candidate oncogenes might have a cumulative effect on MM cell proliferation.

#### 3.3.3. Effect of Candidate Oncogenes on Survival of Canine MM

Recent studies have pointed out that complex clustered SVs in human MM or specific recurrent SVs (i.e., focal amplifications of CFA30) in canine MM are linked to poor outcomes [15,24]. To explore the impact of the SV profiles on canine MM, the SV profiles were determined using low-pass sequencing on 42 canine MM FFPE samples with available survival data (part of a previously published cohort by Prouteau et al.) [24]. In this cohort, the minimal region of CFA 30 amplification was delimited between 16.2 Mb and 16.8 Mb. This region was amplified in 64.3% of cases and contained *GABPB1*, *USP8*, *TRPM7*, and *SPPL2A* genes. While high SV content in canine MM is associated with a poor outcome (one-sided log-rank *p*-value = 0.025), a high SV is less significantly associated with a poor outcome than is the amplification of CFA30 candidate oncogenes (one-sided log-rank *p*-value = 0.0012) (Figure A7).

### 3.4. Exploring the Value of CFA 30 Oncogenes in Human Melanomas

The canine model of MM allowed us to identify new oncogenes; thus, we explored the involvement of these canine CFA 30 oncogenes in human cancers, more specifically, in melanomas, according to published data. Since *TRPM7* and *USP8* CRISPR inactivation has an impact on the majority of human cell lines (703/990 and 960/990 for TRPM7 and USP8, respectively), these genes are considered to be common essential genes (https://depmap.org/, accessed on 1 July 2021). Moreover, a high expression of TRPM7, USP8, GABPB1, or SPPL2A was shown to be associated with shorter survival times in cutaneous melanomas (Log-rank TRPM7, *p*-value = 0.005; Log-rank USP8, *p*-value = 0.017; Log-rank GABPB1, *p*-value = 0.019; and Log-rank GABPB1, *p*-value = 0.022) (https://www.proteinatlas.org, accessed on 1 July 2021). Interestingly, *USP8* knockdown suppressed cell growth, survival, and migration in cutaneous melanoma [75]. In addition, *TRPM7* expression levels have been shown to be associated with an invasive behavior and metastatic potential in cutaneous melanoma cell line [76] and is also expected to act as a protector in both melanocyte physiology and in melanoma cells [77].

Human MM and acral melanomas are characterized by a higher SV content and a lower tumor mutational burden than UV-induced cutaneous melanomas; thus, they are more likely to present strong focal amplifications of the corresponding genes. In the Cancer Genome Atlas, we found one case of acral melanoma (TCGA-ER-A19T-01) harboring a strong focal amplification on chromosome 15 (orthologous to canine chromosome 30) encompassing the candidate oncogenes *SPPL2A*, *USP8*, *TRPM7*, and *GABPB1*. This amplification of 29 copies of these genes was associated with high gene overexpression (16.8-, 22.7-, and 30.2-fold for *TRPM7*, *USP8*, and *SPPL2A,* respectively). In another cohort of 34 patients with acral melanomas, we found one tumor with an intrachromosomal rearrangement on chromosome 15 targeting the genes *TRPM7* and *MYO5A* [78]. Furthermore, two recent studies involving MM cases that performed WGS on 65 and 67 tumor samples showed that chromosome 15 frequently had a deletion in the proximal part, and more rarely, amplifications targeting the candidate orthologous region identified in the present study [14,15]. Moreover, in a cohort of 65 patients with only oral melanoma, *TRPM7* was one of the 48 significantly mutated genes [15].

To confirm the importance of these genes in non-UV-induced melanomas in humans, we evaluated the proliferation capability of two human melanoma cell lines, one mucosal melanoma cell line (HMV-2/CVCL_1282), and one acral melanoma cell line (WM3211/CVCL_6797). Using siRNA to induce silencing of the *TRPM7*, *SPPL2A*, and *GABPB1* genes, we showed that their expression was decreased by 61–69% (Appendix A), which also led to a significant decrease in cell proliferation from 10.8–27.4% (Figure A8). 

These findings confirm that the candidate genes of the canine CFA30 16.2–16.8 Mb region/Human CFA15 50.3–50.9 Mb region are also involved in a subset of human non-UV-induced mucosal and acral melanomas. The results of this study suggest that the therapeutic target potential of these genes in human MM, especially those of the oral cavity, should be explored.

## 4. Discussion

In the last decade, dog models have been established as relevant models for clinical and genetic studies of human MM [13,23,24,79,80,81]. The present work provides a better understanding of the genomic and transcriptomic profiles associated with canine MM by identifying two molecular subgroups of canine patients differing in their transcriptomic profiles and SV content. Thus, these findings suggest different tumor microenvironments for each subgroup, each requiring different types of therapies. In addition, this work allowed the identification of new candidate oncogenes for MM, which should be of interest to human oncology.

The first molecular subgroup contained a majority of tumors with a relatively low SV content, with overexpression of genes related to the microenvironment, particularly the T-cell response and cytotoxic functions. Thus, the likely presence of an effective T-cell infiltrate in tumors in this group, in addition to the overexpression of immune checkpoint proteins, such as CTLA-4, TIM3, and LAG3, favors a response to immunotherapy, especially when using checkpoint inhibitors [52,53,54,82]. Interestingly, cell-type-specific enrichment analysis from our MM bulk RNAseq data with the xCell program [83] confirmed that group 1 samples contained an enrichment of immune cells, such as dendritic cells, macrophages M1 and lymphocytes T (Figure A9). While several studies have linked high tumor mutational burden to a better response of immunotherapy [62,84,85], two recent publications have shown that CNA/SV levels are also predictive of a response to immunotherapy in different cancer types. Among these cancers, cutaneous melanomas harboring a high level of CNAs have the poorest response rate to immunotherapy [52,53]. Ock et al. also suggested that these alterations (CNA/SV) are stronger predictors of the response to immune checkpoints than tumor mutational burden [53]. Following these studies, we suggest that the first molecular subgroup with low SV identified in this study may respond to immunotherapy, while the other subgroup with higher SV rates may respond better to targeted therapies (e.g., anti-CDK4). This hypothesis is further reinforced by the results of a previous study by Ock et al., who observed that the differentially expressed genes between the two groups were enriched in genes associated with response to immunotherapy in human cancers according to [53]. We found similarities in terms of the expression modifications (up or downregulation) in the genes differing in expression between the two canine MM subgroups and the immunotherapy responders and non-responders in the study by Ock et al. (*p* = 1.0 × 10^−11^) [53] (Appendix A). However, several genes associated with the acquisition of an invasive, dedifferentiated, EMT-like phenotype were also overexpressed in this group. While a favorable tumor immune microenvironment is critical for effective immunotherapy, tumor cells could exploit the dedifferentiation program to resist immunotherapy. Further studies combining therapeutic trials with single-cell RNAseq would be very relevant to better assess the benefit of immunotherapies in this group.

In a recent study, Newell et al. also identified two distinct subgroups of human MM based on the degree of localized complex rearrangements [14]. While, here, we classified MM using a transcriptomic-based strategy, we confirmed the existence of two MM subgroups with different SV profiles targeting similar driver genes (*MDM2*, *CDK4*). The association of different SV profiles between the two molecular subgroups of MM could reflect the involvement of different DNA repair mechanism alterations and, thus, different tumorigenesis pathways. The second subgroup was characterized by the presence of recurrent focal high amplifications and numerous chromosomal rearrangements (“high SV”). This chromosomal instability has already been described in human melanomas and is associated with poor outcomes in cutaneous and oral melanomas [15,86]. This aggressiveness was also observed in canine oral melanoma harboring focal amplification on CFA 30 (median survival time of 159 days vs. 317 days for MM with CFA 30 amplification or no CFA30 amplification, respectively, *p* = 5 × 10^−5^) [24], and we also demonstrated that dogs with MM carrying a high SV burden had a poorer prognosis (one-sided *p* = 0.0255) than those carrying a lower SV burden. While the molecular subgroup with high SV exhibits a higher mitotic index (mean = 21.8 mitosis vs. 9.8 for the high-SV group and the low-SV group, respectively, Mann–Whitney test, *p*-value = 0.022), further studies, including data on clinical staging, are needed to refine the correlation between the molecular subgroups and known clinical/histological prognostic factors.

Focal amplification in melanomas results from complex genomic catastrophes previously described as chromothripsis in humans [14,15,86] and dogs [29]; however, most events are too complex to confidently assign to one particular type of mutational mechanism [14]. The WGS of four canine MM cell lines showed that the focal amplifications had complex rearrangements reminiscent of chromothripsis, with chained or clustered breakpoints localized to a subset of chromosomes in regions that also contained copy number oscillations. The formation of double minutes is expected in chromothripsis to explain the high amplification of driver oncogenes. The FISH analysis in this study revealed large, homogeneously stained derivative chromosomes from CFA 10 and/or CFA 30, but no double-minute chromosomes. Similar patterns of high *MDM2*/*CDK4* amplification with large derivative chromosomes were observed in human glioblastoma, and were assigned to chromothripsis, with aggregation of double minute chromosomes [66]. In human acute myeloid leukemia, such catastrophic events have been linked to somatic *TP53* mutations and, thus, p53 dysfunction [87]. In dogs, approximately 50% of MM harbor an amplification of the *TP53* inhibitor, MDM2 [24], and this alteration is mutually exclusive with *TP53* mutations [29]. Similarly, inactivated p53 mutations [11,12,13,14,15,17,88,89,90,91] and *MDM2* amplifications [13,14,15,16] have been observed in human mucosal and acral melanoma, suggesting that p53 pathway dysregulation may be crucial in non-UV-induced melanoma development [29]. 

In human melanomas, recurrent focal amplifications are frequent (over 50%) and significantly less common in cutaneous melanomas. The focal amplification of 5p15 (*TERT* gene locus) is mainly observed in oral MM [92]; recurrent interchromosomal translocations between HSA 5 (*TERT*) and HSA 12 (*CDK4*) were observed in 39.6% of cases [15]. Similarly, over 50% of canine oral MM cases harbored recurrent focal amplification of *CDK4/MDM2* oncogenes. However, unlike human MM, these amplifications did not co-occur with *TERT* DNA amplification, even though TERT RNA was significantly over-expressed in the “high SV” cases (log2ratio = 2.54, p-corrected = 6.7 × 10^−3^). These results suggest that focal amplifications of the *CDK4/MDM2* oncogenes in human and canine MM drive the same oncogenic pathway activation by distinct mechanisms. Promoter mutations enhance *TERT* expression in human cutaneous melanoma [93], and these mutations create a de novo binding site for the transcription factor *GABP* heterotetramer activating *TERT* transcription [94,95]. In dogs, *GABPB1* is overexpressed due to CFA 30 amplification; no *TERT* promoter mutations were observed in the four canine cell lines characterized by WGS. Similarly, Hendricks et al. (2018) did not find any *TERT* mutations in five canine MM cases analyzed using WGS to explain *TERT* overexpression. Other mechanisms could be involved in gene overexpression, such as epigenetic regulation or activation of the NFKB or JAK/STAT pathways [93,96,97].

Regarding the focal amplification of CFA 30 (16–17 Mb region), the corresponding amplification of HSA 15 has already been observed in few human melanomas [78,86] (https://www.cancer.gov/tcga, accessed on 1 July 2021), and was associated with overexpression of the amplified genes (https://www.cancer.gov/tcga, accessed on 1 July 2021). Nevertheless, based on the high recurrence of this region amplifications in canine MM (~50% [24]), the CFA 30 region should contain important oncogenes for non-UV-induced melanoma development. The following lines of evidence support this hypothesis: (i) in dogs, the degree of expression of candidate oncogenes of CFA 30 is correlated with the copy number; (ii) these genes appear to be essential in a majority of human cell lines (*TRPM7* and *USP8*); and (iii) they are associated with a shorter survival time in cutaneous melanomas. In the present study, we showed that *TRPM7*, *SPPL2A*, and *GABPB1* are involved in the proliferation of canine and human non-UV-induced melanomas. We hypothesized that genes included in the CFA 30 amplification have a cumulative effect on the tumoral transformation of melanocytes, and further studies are required to explore their roles, such as in tumor proliferation, adhesion, and migration [98]. Interestingly, a recent study using transcriptomic profiling of canine cancers identified *TRPM7* as a biomarker of melanoma and *SPPL2A* as a highly relevant target [99]. *TRPM7* thus appears to be an important gene for non-UV-induced melanomas, as it was recently found to be (i) involved in several other human cancers [98]; (ii) significantly mutated in human oral melanomas [15]; and (iii) involved in intrachromosomal translocation in acral melanoma in a previous study [78]. The importance of these genes for UV-induced melanomas has to be further explored by silencing or inactivating their expression in cutaneous cell lines in order to assess whether these genes could also be relevant therapeutic targets in UV-induced melanomas.

## 5. Conclusions

Domestic dogs are increasingly being considered as a genetic system for the study of human cancers with underlying genetic components, owing to the strong breed-related predisposition to cancers. In this study, we identified two molecular subgroups of canine oral MM, differing in their SV content and gene expression profiles (immune microenvironment vs. pigmentation pathway and oncogenes), which may correspond to different therapeutic options. Moreover, this model allowed us to identify several oncogenes that are relevant for MM development, particularly *TRPM7,* which was recently shown to be involved in rare human non-UV-induced melanomas. These results suggest that it is important to permit the sub-setting of tumors to improve tumor stratification for clinical trials, which will likely have differing clinical outcomes, and to provide the optimum therapy depending on this classification for both canine and human medicine.

## Figures and Tables

**Figure 1 cancers-14-00276-f001:**
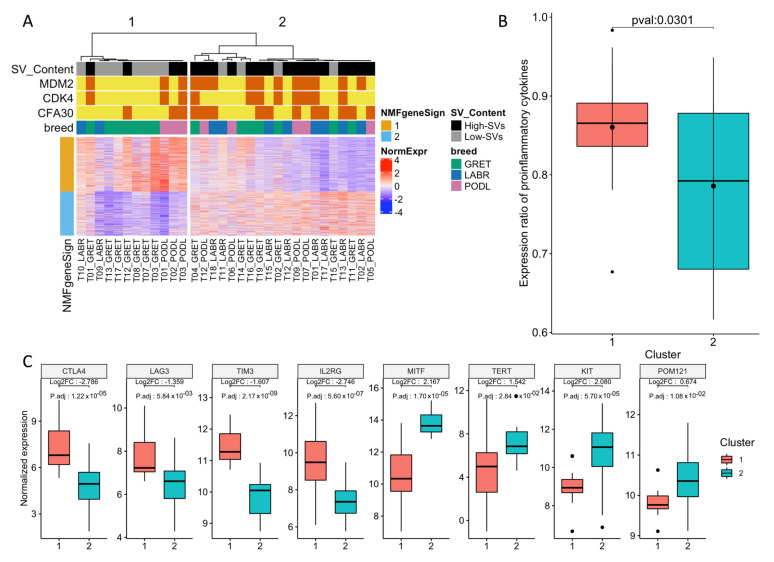
Transcriptomic analysis of 32 canine MM samples. (**A**) NMF clustering of expression data identified two subgroups that overlap with SV content. The first subgroup is characterized by the expression of immune response and cytokine-mediated signaling pathways and a low SV content, whereas the second subgroup is characterized by the expression of melanin metabolic processes and pigmentation pathways and is enriched in SV as well as focal amplifications of MDM2, CDK4, or CFA30:17Mb region. (**B**) Expression ratio of proinflammatory cytokines (IFN-γ, IL-1A, IL-1B, and IL-2) and immunosuppressive molecules (IL-10, IL-11, and TGFB1) significantly differs between the two groups (*p* = 0.03, Student test). (**C**) Expression of immune checkpoint genes or known oncogenes according to the transcriptomic classification. The following immune checkpoint genes CTLA4, LAG3, TIM3, and IL2RG are overexpressed in group 1, while the oncogenes MITF, TERT, KIT, and POM121 are overexpressed in group 2.

**Figure 2 cancers-14-00276-f002:**
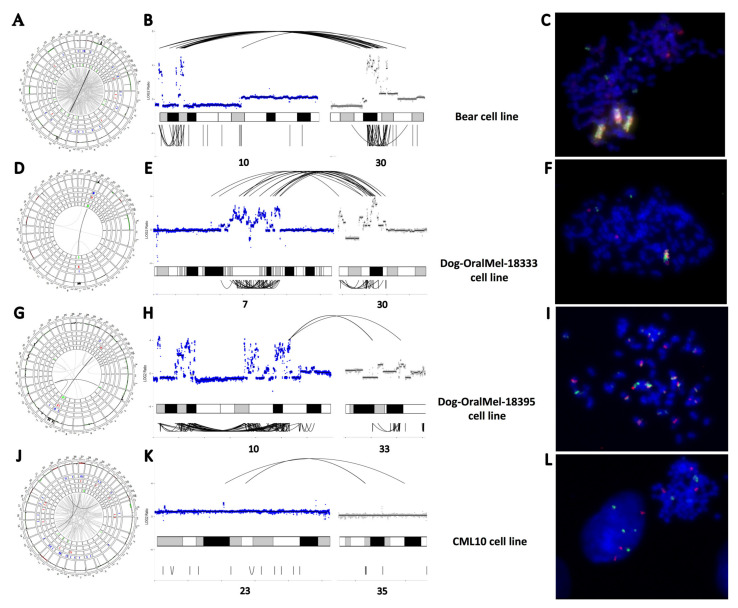
Structural variants (SVs) of the four canine MM cell lines. (**A**–**C**) SVs identified through WGS in the cell line Bear. (**A**). Circos plot representing the distribution of SVs along dog chromosomes with, from external to internal layers, CNA gains/losses (in dark red/green), deletions, duplications, insertions, and inversions in blue, light red, orange, and green, respectively. Interchromosomal break-ends (BND) are represented by gray lines connecting chromosomes with a color intensity corresponding to the number of reads validating the SV. (**B**). Focus on CFA 10 and CFA 30 present the focal amplifications and clusters of SVs (BND and INV). Copy numbers in log2ratio are represented as inter (top) and intra-chromosomal (bottom) SVs. (**C**). Fluorescence in situ hybridization (FISH) analysis of Bear cell line targeting CFA 10 (MDM2 region) and CFA 30 (TRPM7 region) in green and red, respectively, showing derivative chromosomes compatible with a chromothripsis-like event. (**D**–**F**). SVs of the cell line Dog-OralMel-18333. (**D**). Circos plot representing CNV, BND, DEL, and INV across the genome. (**E**). Focus on CFA 7 and CFA 30 presenting the focal amplifications and cluster of SVs (BND, DEL, DUP and INV). (**F**). FISH analysis of Dog-OralMel-18333 cell line targeting CFA 7 and CFA 30 (TRPM7 region) in green and red, respectively, showing derivative chromosomes compatible with a chromothripsis-like event. (**G**–**I**). SVs of the cell line Dog-OralMel-18395. (**G**). Circos plot representing CNV, BND, DEL, and INV across the genome. (**H**). Focus on CFA 10 and CFA 33 presenting the focal amplifications and cluster of SVs (BND, DEL, DUP, and INV). (**I**). FISH analysis of Dog-OralMel-18395 cell line targeting CFA 10 (MDM2 region) and CFA 10 (CDK4 region) in green and red, respectively, showing derivative chromosomes compatible with a chromothripsis-like event. (**J**–**L**). SVs of the cell line CML10. (**J**). Circos plot representing CNV, BND, DEL, and INV across the genome. (**K**). Focus on CFA 23 and CFA 35 presenting interchromosomal rearrangement. (**L**). FISH analysis of CML10 cell line targeting CFA 10 (MDM2 region) and CFA 30 (TRPM7 region) in green and red, respectively, showing 3 to 4 spots compatible with a tetraploid state without massive genomic rearrangements. Image (**A**,**D**,**G**,**J**) are available in the Appendix A (Figure A3, Figure A4, Figure A5 and Figure A6).

**Figure 3 cancers-14-00276-f003:**
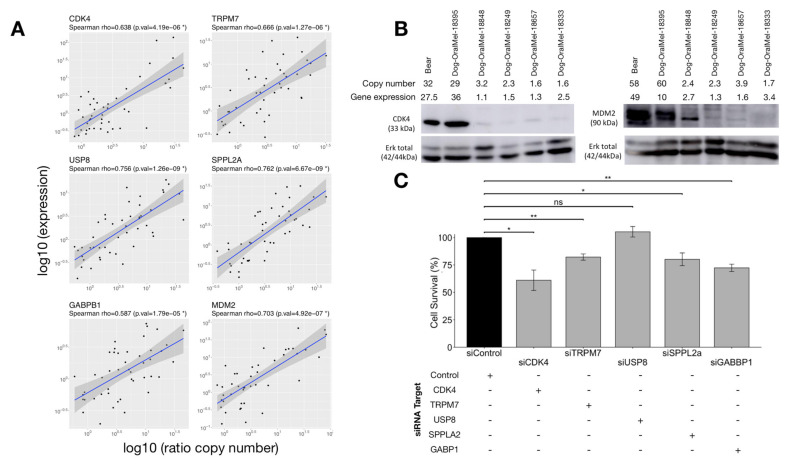
Expression data of the CFA 10 and CFA30 candidate genes at mRNA and protein levels and impact on cell proliferation. (**A**). For the CFA 10 genes (MDM2, CDK4) and CFA 30 genes (GABPB1, TRPM7, SPPL2A and USP8), expression was defined by RT-qPCR and was significantly correlated to copy number in tumor tissue (*n* = 46). (**B**). Western blot analysis in six canine MM cell lines showing a higher amount of protein in cell lines having high copy numbers of CDK4 and MDM2. (**C**). Cell proliferation assay (Bear cell line) showing the effect of the knockdown (siRNA) of candidate oncogenes from CFA 30 and CDK4 on cell survival. The knockdown of CDK4, TRPM7, GABPB1, and SPPL2A had an impact on cell proliferation in comparison with the control siRNA (*t*-test, with * corresponding to *p*-values <0.05 and ** to *p*-values <0.01).

## Data Availability

The RNASeq data are available in SRA under the BioProject PRJNA749900. WGS of canine cell lines are available at PRJNA779870 and low-pass FFPE fastq files are available under the SRA bioproject PRJNA780881. WES fastq files are available in the SRA under the bioproject PRJNA786469.

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
