# Peer review of "Canine Oral Melanoma Genomic and Transcriptomic Study Defines Two Molecular Subgroups with Different Therapeutical Targets"

_cancers, 2022, doi:10.3390/cancers14020276_

Round 1
Reviewer 1 Report
The study by Prouteau et al used transcriptomic and genomic analysis of 32 canine mucosal melanoma (MM) samples to characterise mucosal melanoma into two distinct molecular groups, each differing in their microenvironmental and structural variant content. This study further suggests that characterisation of these subgroups may allow tailoring of specific therapies. The strengths of this study is the focus on a rare melanoma subtype that is understudied and their comprehensive characterisation using genomic and transcriptomic analysis, as well as functional validation of their candidate genes using appropriate cell lines. However, there are some minor comments and queries that could further improve this study.
Can the authors comment on how the two molecular subgroups identified in this study compare to those previously identified based on SV content (as published by Newell, Nat Comms, 2019).
As the authors mentioned, the ratio of immune cell type is important, and in line with this, the authors should attempt to deconvolute their transcriptome data to interpret the relative frequency of pro vs anti-tumor immune cell types using CIBERSORT. It would be worthwhile to compare the types of immune cells between the two molecular subtypes.
Although group 1 shows highly inflammatory features that suggest this subgroup may be susceptible to immune-mediated therapies, they also show features of dedifferentiation. Given that dedifferentiation is associated with resistance to immunotherapies, could the authors discuss how immune therapies may affect this subgroup in this context?
For the four canine cell lines included in the WGS and functional study (Figure 2), could the authors specify which of the two molecular subgroups these cell lines are classified as? This should be considered as part of their analysis and interpretation of data.
The oncogenes identified in the CFA 30 region, could the authors speculate whether these genes are important exclusively in MM melanoma or in other melanoma subtype such as cutaneous melanoma. It would be interested to compare whether siRNA silencing of the TRPM7, SPPL2A, and GABPB1 genes have similar or distinct effects in cutaneous melanoma cell lines to determine whether effects of these genes are exclusive to mucosal melanoma.
Reviewer 2 Report
The research question addressed in the manuscript is a prominent issue within the oncology field and the results and discussion greatly contribute to a type of neoplasm that is difficult to therapeutically control. In addition, the cutting edge and broad molecular approach that was used in this study can contribute to pave the way for the knowledge of comparative oncology, which promises to seek solutions with a translational profile.
I congratulate the researchers for conducting this research and the article presented is very well written and presents solidity in the data that were described in a very organized way.
I have few notes that do not compromise the quality of the work presented.
1-since the authors used samples for dogs and not only melanoma cell lines it would be important to explain (or consider) when they reinforce the hypothesis that the first molecular subgroup with low SV may respond to immunotherapy (predictive factor) and dogs with MM carrying a high SV burden had a poorer prognosis if the samples analyzed came from patients with same clinical stage or not. This information is important since it is known that clinical staging including lymph node infiltration, and mitotic index are a relevant prognostics factors for canine oral melanoma. Do results show which molecular subgroup is correlated with these clinical or histological features? Or doesn’t exist? Could be included at discussion topic, on line 593.
2- Figure S3 is repeated twice, and the legend (superscription) is not very readable (increase font size).
